# Cybersecurity Testing for Automotive Domain: A Survey

**DOI:** 10.3390/s22239211

**Published:** 2022-11-26

**Authors:** Feng Luo, Xuan Zhang, Zhenyu Yang, Yifan Jiang, Jiajia Wang, Mingzhi Wu, Wanqiang Feng

**Affiliations:** 1School of Automotive Studies, Tongji University, Shanghai 201804, China; 2Nanchang Automotive Institute of Intelligence and New Energy, Tongji University (NAIT), Nanchang 330052, China

**Keywords:** automotive, cybersecurity testing, penetration testing, fuzzing, model-based testing

## Abstract

Modern vehicles are more complex and interconnected than ever before, which also means that attack surfaces for vehicles have increased significantly. Malicious cyberattacks will not only exploit personal privacy and property, but also affect the functional safety of electrical/electronic (E/E) safety-critical systems by controlling the driving functionality, which is life-threatening. Therefore, it is necessary to conduct cybersecurity testing on vehicles to reveal and address relevant security threats and vulnerabilities. Cybersecurity standards and regulations issued in recent years, such as ISO/SAE 21434 and UNECE WP.29 regulations (R155 and R156), also emphasize the indispensability of cybersecurity verification and validation in the development lifecycle but lack specific technical details. Thus, this paper conducts a systematic and comprehensive review of the research and practice in the field of automotive cybersecurity testing, which can provide reference and advice for automotive security researchers and testers. We classify and discuss the security testing methods and testbeds in automotive engineering. Furthermore, we identify gaps and limitations in existing research and point out future challenges.

## 1. Introduction

With the development of information and communications technology (ICT), automobiles have gradually become intelligent and interconnected. While increased connectivity has brought us convenience and comfort, it has also made automobiles more vulnerable to external cyberattacks [1]. Over the past decade, frequent automotive cybersecurity incidents have exposed cybersecurity vulnerabilities in automobiles. In particular, Miller and Valasek’s compromise of Jeep led to the recall of 1.4 million vehicles, which brought public attention to automotive cybersecurity [2]. Therefore, it is crucial to effectively test and discover threats and vulnerabilities before vehicle production. Much of the research on automotive cybersecurity has focused on security analysis [3] and countermeasures [4]. This positive direction can expose many security issues. However, the issues may not be fully considered, and potential vulnerabilities will be overlooked. Several studies highlight the urgent need for security testing in the current automotive industry [5]. Security testing is an effective way to identify security vulnerabilities and is a practical activity to verify the security requirement of the system under test (SUT). In addition, with the release of automotive cybersecurity standards and regulations, such as SAE J3061 [6], ISO/SAE 21434 [7], and WP.29 R155 [8], the importance of security testing has also been emphasized, and security testing is an essential part of the automotive security development life cycle [9]. These regulations and standards state that security testing is imminent but do not provide OEMs and suppliers with specific implementation methods. The only high-level guidance makes the actual execution of the test difficult. Due to the complexity of automotive systems, various hardware and software technologies are involved and the sources of threat vulnerabilities are diverse. Thus, automotive security testing is very challenging. Currently, it is unclear what cybersecurity testing methods are available in the automotive field and the extent to which each method is utilized. Moreover, the cost, automation level, and safety of security testing are also issues encountered in the testing process. These problems are the starting points for our study.

To the best of our knowledge, there are currently few review articles on automotive cybersecurity testing. Pekaric et al. [10] perform a systematic mapping study (SMS) on automotive security testing techniques. They discuss various security testing techniques and map them to the vehicle lifecycle, the AUTomotive Open System ARchitecture (AUTOSAR) layers, and attack types. However, they study security testing from a macro perspective and do not cover testing technical details and applications. Mahmood et al. [11] outline a few testing methods and testbeds, but the categories are not comprehensive. There is no comparative analysis of the characteristics of the various techniques. Bayer et al. [12] provide a short description of embedded security assessment, which includes practical security testing. Security testing is divided into four types: functional testing, vulnerability scanning, fuzzing, and penetration testing. However, there are only simple usage scenario descriptions, which also lack comparative analysis. Ebert et al. [13] introduce security testing techniques from the perspective of tools, outline the advantages and limitations of widely used testing tools, and compare their scalability, usability, and availability. Their research aspects only focus on testing tools, and the analysis dimension is one-sided. Due to the limitations of previous surveys, we expect to complete a more comprehensive review of automotive cybersecurity testing through a systematic literature search process.

As far as we know, this is the first systematic and complete review of existing automotive cybersecurity testing with a systematic literature review (SLR) approach. Since 2022 is still ongoing, and we collected literature by year, some relevant publications may be published at the end of the year. These publications may affect the statistical results; therefore, we only screened the literature up to 2021. We thoroughly describe the different testing methods and testbeds used in the automotive field from 2010 to 2021. This paper shows the technical details and application scenarios of various technologies by comparing the literature. Through the research presented in this paper, we illustrate the importance of automotive security testing and demonstrate some existing methods and testbeds, including their characteristics and applications. This paper provides a concrete foundation and guidance for engineers and researchers in this research area. We expect this work can help them to better evaluate automotive cybersecurity with practical tests, rather than being limited to theoretical analysis and design.

The rest of this article is organized as follows: Section 2 presents the process of conducting a SLR. Section 3 introduces the methods of automotive cybersecurity testing. Section 4 compares the characteristics of different cybersecurity testing testbeds. The research challenge of the target topic is analyzed in Section 5. Finally, a conclusion is drawn in Section 6.

## 2. Methodology

### 2.1. Research Questions

The main goal is to conduct a more comprehensive review of cybersecurity testing in the automotive domain. The paper summarizes the state of the arts and challenges in the topic area. This section raises the following three research questions (RQs) addressed in this paper:RQ1. What are cybersecurity testing methods applied in the automotive domain?RQ2. What testbeds are used for automotive cybersecurity testing?RQ3. What are the research challenges in this topic area?

These three RQs address the issues raised in Section 1. RQ1 focuses on the appropriate methods of cybersecurity testing utilized in the automotive domain. Because of the cost and safety of real vehicle testing, security testbeds are used. RQ2 aims to research automotive cybersecurity testbeds from 2010 to 2021. RQ3 discusses challenges for automotive cybersecurity testing, including current limitations and future trends.

### 2.2. Search Process

We collected and analyzed the literature in a manual search method. The search process was conducted based on the following steps.

#### 2.2.1. Search Database

Six common databases were adopted, including SAE Mobilus, commonly used in the automotive industry. Additionally, the latest research work on security is generally presented at security-related conferences, such as Black Hat, Defcon, etc. Therefore, Google Scholar was used as a supplement in the backward snowballing phase. Table 1 shows the databases for the search.

#### 2.2.2. Search String

We needed to construct a search string containing boolean operators to search the relevant literature. The search string needed to be adjusted for different databases. The general search string can be determined as follows.

(automotive **OR** vehicle) **AND** (security **OR** threat **OR** vulnerability **OR** risk) **AND** (fuzz **OR** penetration **OR** test)

We focused on the literature related to automotive cybersecurity testing; therefore, the first bracket in the search string indicates that it is related to vehicles, the second represents security, including threats, vulnerabilities, and risks, and the third shows that it is related to testing. Of course, security testing also includes fuzz and penetration.

#### 2.2.3. Search Procedure

The first step of the search process was to filter the literature in seven databases using the defined search string. The second was to screen the literature based on inclusion and exclusion criteria. Then, we refined the literature through abstracts and full texts. Finally, we identified new papers through backward snowballing, a method for including new papers by examining the title, publication venue and authors in the reference list of the selected paper [14]. Meanwhile, we used Google Scholar to supplement the search for some security conference articles. Figure 1 shows an overview of the entire search process.

### 2.3. Search Criterion

After the initial screening of the search string, it was necessary to determine the search criteria for further selection. The search criterion can be seen in Table 2.

### 2.4. Search Results

The search process can be seen in Table 3. We calculated statistics based on the final selected papers and plotted a graph from three aspects: year of publication, and number and source of papers. Figure 2 depicts the distribution of papers by source from 2010 to 2021. We can see a significant increase in articles on the target topic starting in 2015 due to Miller and Valasek’s remote control of a Jeep that led to the recall of 1.4 million vehicles, drawing public attention to automotive cybersecurity.

## 3. Automotive Cybersecurity Testing Methods

Concerning RQ1, as shown in Figure 3, cybersecurity testing methods can be classified from three perspectives: knowledge level, level of automation, and test objective. The knowledge-based method focuses on the knowledge level of the SUT. Depending on the knowledge level, it can be divided into three types: black-box, white-box, and gray-box tests. The automation-based method pays attention to the level of automation of the test tool. It can be classified into three types based on the level of automation of the tool: fully automated testing, semi-automated testing, and manual testing. Based on the test objective, it can be classified into threat-based and requirement-based types. The threat-based method conducts testing to reveal threats and vulnerabilities of the SUT. This method aims to evaluate the security of the target system. Threat-based testing can be categorized into four sub-types: vulnerability scanning, penetration testing, fuzzing, and risk-based testing. Requirement-based testing is performed to verify the security functional requirements and specifications. Thus, we also refer to it as functional security testing. Moreover, model-based testing has also been discussed as an advanced testing technique in this paper.

### 3.1. Knowledge-Based Testing

The knowledge-based testing is conducted based on the degree of knowledge of the SUT. In general, it can be divided into the following three categories.

For black-box testing, testers do not have functional specifications and documents related to the SUT, and the target system is treated as a black box. Testers verify the security design and defense of the system from the outside, which is closer to the actual attack situation and can assess the system’s resistance to attacks. Since the information is not available, it can be challenging in the early stage of testing, requiring significant time and cost to reverse engineer the system. Moreover, without specifics, potential threats and vulnerabilities are difficult to identify. Penetration testing is a kind of black-box testing in most cases and will be thoroughly presented in Section 3.3.2.

White-box testing means that the internal details of the target system are known. Potential threat vulnerabilities can be carefully revealed based on functional specifications and source code without wasting much time and effort on information acquisition. Test coverage can be significantly improved by accessing the full source code, but this would take a significant amount of time. This method is commonly used in dynamic and static code scanning of automotive applications [15]. This approach enables comprehensive testing of the automotive system from source code to architecture design.

Grey-box testing is a combination of black-box and white-box, a method that takes into account time and cost trade-offs. Grey-box testing means that the tester knows part of the information of the SUT and can build test cases based on acquired knowledge, which takes less time and effort than white-box testing. Ebert et al. [16] proposed a grey-box testing method. It can obtain part of the knowledge in the system and determine the high-priority risks to test through security analysis, thereby saving testing time and resources. Its good traceability is also beneficial for validating our testing requirements. A comparison of the three methods is given (see Table 4).

### 3.2. Automation-Based Testing

Automated-based testing can be classified into three types based on the level of automation of testing tools and frameworks: fully automated, semi-automated, and manual testing.

With the increasing complexity of electronic and electrical systems in automobiles, manual testing has been unable to meet the growing testing requirements. Automotive development is a cost-sensitive activity, and the time and cost of testing must be considered throughout the vehicle development lifecycle. Automated testing can significantly improve test efficiency, reduce human workload, and avoid deviations caused by human subjective factors. For example, in the security audit of the source code, if the security testing of the software is performed by manual testing, it is time-consuming and complicated. In this case, an automated code inspection tool will be commonly utilized. Testing tools may only be dedicated to some specific application scenarios and cannot fully automate the entire testing process. It is necessary to cooperate with some automated testing frameworks or testbeds to form a complete testing process. A fully automated method is ideal and expensive, so it is usually superior to use a semi-automated testing method that combines manual and automated testing. In our surveyed literature, Marksteiner et al. [17] proposed a conceptual framework to transfer threats into executable test cases using a workflow-based system. In 2021, they also presented an automated automotive cybersecurity security testing process based on ISO/SAE 21343, bridging the gap between existing automotive security standards and actual system testing [18]. Table 5 shows the comparison of the automated testing methods.

### 3.3. Threat-Based Testing

Threat-based testing is a threat-centric approach. Security threats can lead to vulnerabilities in automotive safety-critical systems that attackers can exploit to cause catastrophic consequences. Therefore, it is essential to find the potential threats to the system. The threat-based method aims to discover loopholes in the system, identify potential attack paths and exploit the vulnerabilities to attack the system. Several threat-based testing methods are discussed below.

#### 3.3.1. Vulnerability Scanning

Vulnerability scanning is a typical security testing approach used in the traditional IT industry. It generally refers to using automated scanning tools or scripts to discover potential vulnerabilities of target systems based on common security vulnerability databases, such as common vulnerabilities and exposures (CVE), national vulnerability database (NVD), etc. For source code scanning, it must also comply with the corresponding code specifications, such as MISRA C [19]. This method can quickly and efficiently expose some security risks, especially those known in the database. However, the vulnerabilities discovered in this way are not comprehensive, and some unknown attacks may require further identification with other testing technologies. Vulnerability scanning is a more proactive security measure to detect threats than passive defenses such as automotive firewalls. The cybersecurity architecture of ICVs can be described in three aspects: cloud, pipe end, and vehicle end. For cloud infrastructures, traditional IT-based vulnerability scanning can be used to discover the open ports and services of the backend server. This section is devoted to two other vulnerability scanning approaches in the automotive domain. It can be classified into static/dynamic code analysis and communication service scanning.

The first one is static/dynamic code analysis. The software-defined vehicles (SDV) concept has recently become popular. The software has become the basis of future automotive intelligence. Today, the software size in cars has exceeded 100 million lines of code [20], and software code security has become a critical task. Objects for static/dynamic analysis can be source code or binary files. Testers can use automated tools to find known vulnerabilities, such as stack overflow [21]. Alternatively, they also can use lightweight scripts to check whether the program’s system security hardening measures are enabled, such as data execution prevention (DEP), position independent executable (PIE), etc.

The other is communication service scanning. The communication here can be wireless or wired communication. Traditional vulnerability scanning can be used for classic wireless communication, such as Wi-Fi, cellular network, and Bluetooth, which is also the most commonly used attack vector for hackers to compromise vehicles [1]. This attack is more straightforward and less expensive than accessing a physical interface like on-board diagnostics (OBD). Franco et al. [22] utilized Nmap and Nessus scanners to obtain sensitive data of car infotainment systems through Wi-Fi, such as a physical address, chip manufacturer, open ports, and conduct a denial of service (DoS) attack on Wi-Fi communication. The other is the physical interface of direct contacts, such as the OBD interface, which generally runs the unified diagnostic services (UDS) on the CAN. UDS is a diagnostic protocol based on the application layer in the open systems interconnection (OSI) model defined by ISO-14229 [23]. Among them, ISO 14229-1 defines diagnostic services. However, it does not involve the network layer and implementation details, only the content of the application layer so that it can be implemented on different automotive buses such as CAN, Local Interconnect Network (LIN), Flexray, Ethernet. and K-line. Attackers can tamper with the configuration of the electronic control unit (ECU) through the UDS protocol, which can affect the functionality of the ECU. However, before reading and writing data, a simple security access verification is required by the UDS security access service. The client must send a seed request first, and then the server replies with the seed of a positive response. The client calculates the secret key according to the seed and the security algorithm. The client sends the secret key, and the server compares the received secret key with its secret key. If the keys match, the server unlocks the relevant data or service. If it does not match, it is considered a bad attempt. There are several security holes in this verification process. First, if the randomness of the seed is weak, the system can be easily broken by an attacker. In addition, if there is no time interval between false attempts, then the attacker can use brute force to crack. Therefore, we can use some automated scanning tools or scripts to verify the randomness of the seed and whether there is a delay among multiple false attempts. In our other work [24], we used our self-developed software tool to automatically discover ECUs with UDS protocol in the in-vehicle network and further identify the UDS services supported by various ECUs. The information obtained from the previous UDS topology discovery and service scanning can facilitate subsequent penetration testing and fuzzing. Weiss et al. [25] also designed a security scanning tool for the CAN transport layer to help assess the attack surface of the system. Table 6 compares various vulnerability scanning methods by time, cost, test scope and limitation.

#### 3.3.2. Penetration Testing

Penetration testing, also called pen testing, is a traditional testing method commonly used in the IT field to test the security of web applications. Security testers usually imitate hackers to conduct malicious attacks on the system under test to discover system security loopholes. The objectives tested generally contains applications, communication networks and security-critical systems. As it develops, penetration testing is not only a testing technology but also a testing process. Nowadays, there have been some agreed rules that describe the steps of penetration testing. One of the most famous is the penetration testing execution standard (PTES). In summary, PTES can be divided into five steps: intelligence gathering, threat modeling, vulnerability analysis, exploitation, and reporting. Intelligence gathering is the first step of penetration testing. This phase is responsible for collecting security-related information about the SUT, such as system architecture, communication methods, etc. This information is beneficial to the subsequent penetration work. The threat modeling phase analyzes the possible threats to the system through threat analysis and risk assessment and determines the best way to attack. Vulnerability analysis combines the information from the first two steps and identifies the attack points. The exploitation phase conducts the actual attack on the identified threat vulnerabilities to reach the purpose of penetration testing. Finally, the report phase summarizes the entire penetration process and condenses the details of the test. Although there is no standardized penetration testing methodology for the automotive sector, the above-mentioned traditional testing process for IT can provide a reference for the application of penetration testing in automobiles.

From SAE J3061 in 2016 to SAE/ISO 21434 in 2021, the standards clearly state that penetration testing should be performed as a necessary activity in the verification and validation phase during automotive development. However, the standard does not guide the specific implementation details of automotive security testing, so it is necessary for us to conduct this study to investigate the application and prospects of penetration testing in the automotive industry by reviewing the research of scholars and experts.

Concerning automotive penetration testing, we analyzed 20 papers and found that penetration testing is one of the most widely used security testing techniques. Generally, automobile penetration testing includes component and vehicle penetration testing. Since various components are sourced from different suppliers, each part must be tested individually before system integration to ensure security. A typical penetration test is black-box, so it depends more on the tester’s experience. From the attack vector’s perspective, the penetration testing objectives can be divided into network communication, software, and hardware.

Automotive network communication is divided into internal and external communication. Internal communication contains wired types such as CAN, LIN, and automotive Ethernet, while external communication includes wireless types such as Wi-Fi, Bluetooth, cellular network, etc. The automotive network is a common entry point for attackers to break into vehicles. Due to possible flaws in the design and implementation of network communication protocols, security testers utilize various attack methods to violate the security properties of network communication from the perspective of attackers, thereby discovering potential security vulnerabilities in automotive network communication. The security attributes involved in communications generally include confidentiality, integrity, authentication, availability, authorization (CIAAA). For each security attribute, different attack methods can be applied, such as sniffing, tampering, spoofing, DoS, unauthorization access, etc. Testers build single-step or multi-step combined attacks that violate the security properties according to the security requirements of real scenarios and evaluate the security risk.

As for software testing, penetration testing mainly performs malicious operations on the application software, such as injection and tampering, to change its control logic. Testers can tamper with the ECU firmware or employ the UDS function to affect the behavior of the ECU [26]. They also can install malicious apps on the smartphone to connect to the car’s in-vehicle infotainment system (IVI), then exploit the IVI vulnerability to send malicious CAN messages that impact the safety-critical function [27]. Through these attacks, the software system’s defense resistance to malicious attacks can be assessed, and the vulnerability of the software can be effectively discovered.

Regarding hardware, the research on attack and testing sensors such as lidar, millimeter-wave radar, and cameras has become increasingly popular in recent years. As the core components of intelligent networked vehicles, sensors play a crucial role in guiding driving. If the sensor is compromised, it could endanger human life. Yan et al. [28] performed spoofing and jamming attacks on multiple Tesla sensors, causing the automotive self-driving to malfunction. Shin et al. [29] present a saturation attack to affect the ability of the lidar to detect objects.

A brief comparison of penetration testing in the surveyed literature from the perspectives of attack vectors, attack types, and knowledge level (■ = Black box, ☐ = White box, 🞕 = Grey box) can be found in Table 7. Figure 4 and Figure 5 show the distribution of papers by attack type and vector. In the surveyed literature, sniffing/eavesdropping and injection attack are the most used, followed by spoofing, and DoS is the third. As for the attack vector, the in-vehicle network is the most commonly used attack entry point for penetration testing as it undertakes critical communication tasks inside the vehicle.

#### 3.3.3. Fuzzing

Fuzz testing, also named fuzzing, is a software testing technique that verifies the security and robustness of the SUT. This approach identifies security vulnerabilities in the SUT by feeding a large amount of random or unexpected data to and monitoring the behavior of the SUT. Fuzzing comprises three parts: a test case generator, a monitoring system, and a test environment. This testing is generally executed by a dedicated software tool called fuzzer. Fuzzer can be divided into two types based on how the test data is generated: mutation-based and generation-based. The mutation-based approach generates test cases by randomly mutating existing data samples. It is a black-box approach. However, the generation-based approach is a white-box approach, which uses the syntax and structure of known protocols or files to generate test cases based on specific rules or models. The generation-based approach has a higher test case pass rate and test coverage than the mutation-based approach, but with a relatively higher cost. Therefore, a gray-box testing technique that combines the advantages of both types is sometimes utilized. Although fuzzing is relatively mature in operating systems and application software, it is still rare in automotive systems [20]. In addition, fuzzers in the traditional IT industry cannot be directly applied to the automotive industry. Therefore, the development and application of fuzzing tools in the automotive field is also an important task.

The applicable scenarios of fuzzing are different for various attack vectors. The first one is network communication. The fuzzing of communication protocols is aimed to identify vulnerabilities in their principles or implementations. The second is application services, and this type is mainly for testing the implementation of application services to prevent malicious exploitation by attackers. The third type targets automotive systems, which aims to discover the security vulnerabilities in automotive systems to guarantee the stability and robustness of security-critical systems.

In network communication, Lee et al. [45] mutated the CAN ID and data fields from sniffed CAN messages. The generated fuzzy CAN data was then injected into the CAN bus via the Bluetooth of the computer, and the behavior of the vehicle was monitored. Thus, the vulnerability of the automotive network is exposed. Fowler et al. [46] developed a custom PC-based fuzzer. A mutation-based approach was utilized for CAN bus experiments on the vehicle simulator and instrumentation components to avoid damage to the vehicle. In 2019, they improved their fuzzer prototype and proposed a systematic fuzzy test scheme. The security properties of the ECU were also tested by fuzzy data injection of the display ECU [47]. Werquin et al. [48] developed a sensor harness to facilitate automated detection of the system’s behavior, which has significant advantages over previous manual testing approaches. Furthermore, the method has been integrated into the open-source security testing tool CaringCaribou.

Most of the above fuzzy test cases for CAN are random variants of CAN-specific data, and the test coverage is not high. Radu et al. [49] created initial seeds for fuzzing using relevant static data from control flow graphs extracted from ECU firmware. This approach can improve the test coverage and is better than the random mutation strategy. Zhang et al. [50] proposed a more novel fuzzing approach to expose the vulnerability of CAN. They developed two fuzzy data generators. One reverses CAN by bit-flip rate (BFR) and performs a mutation operation on the identified signals to prevent data combination explosion. The other uses generative adversarial networks (GAN) in deep learning to learn protocol models and generate test cases. They conducted experimental evaluations on intrusion detection systems (IDS) and live vehicles. The results show that both approaches are more effective than the random mutation approach.

In addition to CAN, Nishimura et al. [51] developed an interface to support CAN FD based on the existing fuzzing tool beSTORM and calculated test execution time parameters to evaluate the usability. Li et al. [52] developed a fuzzer for the automotive Ethernet Scalable service-Oriented MiddlewarE over IP (SOME/IP) protocol, and the fuzzer can enable multiple test processes simultaneously to improve testing efficiency. In addition, valid packet headers can be generated by structural mutation, which can successfully expose the implementation flaws of SOME/IP.

In application services, Bayer et al. [53] implemented four UDS service request message models and performed fuzzer tests on simulated ECUs. The results show that their fuzzer effectively detects faults and standards compliance of automotive ECUs. Patki et al. [54] developed a fuzzing tool for the UDS protocol. It uses a mutation-based approach to create invalid inputs to find hidden vulnerabilities in the automotive environment. They also compared their proposed system with other existing fuzzers (Defensics and extended Peach) to highlight the advantages.

As for automotive systems, Moukahal et al. [55] designed a fuzz testing framework, VulFuzz. The framework uses security metrics to rank the security priority of automotive components and tests the most vulnerable components thoroughly. They used the framework to evaluate an autonomous driving system, OpenPliot, and compared the test results with American fuzzy lop (AFL) and a mutation-based fuzzer. The proposed framework performs better in exposing crashes for the same code coverage. In the same year, they improved their fuzzing framework [56]. The new framework can enhance vulnerability identification and improve branch coverage with lower overhead.

In addition to the fuzzing technology itself, researchers are also studying how to integrate fuzzing into existing automotive security engineering. Vinzenz [57] and Oka [58] both recommend performing fuzzing early in the automotive development life cycle and using the results of fuzzing to enhance other testing activities.

Table 8 presents an overview of fuzzing applied in the automotive domain based on the following five dimensions: attack vector, characteristics, knowledge level (■ = Black box, ☐ = White box, 🞕 = Grey box), type and testing platform. Figure 6 shows the distribution of papers by attack vector in fuzzing literature. It can be seen that CAN is the most used attack vector in fuzzing.

#### 3.3.4. Risk-Based Security Testing

Risk-based security testing (RBST) is a risk-oriented approach. Compared with traditional security testing, it incorporates threat analysis and risk assessment (TARA) techniques and can use the results of risk assessment to optimize the security testing process. Figure 7 illustrates the generic process of risk-based security testing based on the surveyed literature. As the first step, TARA aims to identify vulnerabilities and threats, and then assess and prioritize system security risks. Step 2 refers to model design. This step aims to design a security test model based on a threat model for risk assessment and a functional or behavioral model for system requirements. The goal of step 3, test case generation, is to select the appropriate criteria and algorithms to generate test cases from the test model. Test cases can be automated scripts or manual test scenarios. The test execution runs the generated test cases based on the test environment. Test cases are executed either by automated scripts or manually. The test execution results are often some new threat vulnerabilities that can be fed back into the TARA step as a complement to starting a new round of testing. Thus, RBST is an iterative testing methodology with TARA playing a key role.

TARA serves as the core of risk-based testing. TARA results include threat identification, probability and impact of threat scenarios, and risk values. Using the risk assessment results can help prioritize test cases and test execution. In the past, risk assessment was not an activity in automotive development. However, with the promulgation of SAE J3601 and ISO/SAE 21434, risk-based security activities have been introduced throughout the automotive cybersecurity development process. The standard also provides a TARA methodology for OEMs and suppliers to implement, as shown in Figure 8. Moreover, based on the research literature, Table 9 describes several security analysis methods and techniques in the automotive domain. It also compares the threat models. In addition, whether the analysis method involves safety and security is considered.

### 3.4. Requirements-Based Testing

The requirement-based testing (RBT) is to check whether the technical security requirements of the SUT are correctly implemented. Unlike threat-based testing, which targets threats and vulnerabilities, RBT is essentially the verification of compliance with a requirement specification or standard. The requirements here refer more to the functional requirements of the SUT, so functional security testing will be described below.

#### Functional Security Testing

The goal of threat-based testing is to identify unknown system vulnerabilities caused by security design flaws and code defects, usually using malicious means to attack the system like a hacker. The threat-based testing aims to identify unknown system vulnerabilities caused by security design flaws and code defects. This testing is accomplished by mimicking a hacker using malicious means to attack the system. However, functional security testing is mainly to verify that the security-critical functions or systems are correctly implemented according to the technical requirements in the design specification. Functional security testing is more concerned with the correctness, performance, compliance, and robustness of security-critical systems [12]. It is a kind of white-box testing, which requires a thorough understanding of the technical requirements, implementation, and configuration of security mechanisms. Compared with traditional functional testing, it focuses more on security-related functions, such as encryption/decryption and authentication algorithms, intrusion detection systems, etc.

In order to ensure the security of the internal communication data of the connected car, a secure in-vehicle communication solution is generally adopted. The secure onboard communication (SecOC) in the AUTOSAR software specification is generally acknowledged in the automotive industry. The SecOC mechanism can verify the authenticity, integrity, and freshness of the communication between ECUs. Figure 9 shows the freshness verification and message authentication process of SecOC. On the sender side, the SecOC module uses the secret key to generate the message authentication code (MAC) for the message protocol data unit (PDU). It adds the freshness value to obtain the Secured I-PDU. After receiving the Secured I-PDU, the receiver will use the same algorithm to check whether the MAC of the message is consistent with the sender to ensure the authenticity and integrity of the data. Additionally, the freshness value can prevent replay attacks. However, SecOC cannot guarantee the confidentiality of data, so researchers developed CINNAMON (Confidential, INtegral aNd Authentic on board coMunicatiON), which extended the SecOC module to ensure the confidentiality of transmitted data [68]. It is necessary to know the corresponding algorithms for testing cryptographic and verification functions. If not, it needs to be fetched by the reverse engineer, such as machine learning [69]. After obtaining the algorithm, the inverse operation can be performed in the test system to verify whether the output is consistent with the input.

### 3.5. Model-Based Testing

The above describes the cybersecurity testing methods for automobiles from three perspectives: the level of knowledge of the system, the automation level, and the test objectives. Next, we would like to illustrate an advanced testing technique that may be applied in these testing methods.

Model-based security testing (MBST) is a relatively advanced security testing technology. It is a combination of security testing and model-based testing, mainly through models, to verify the security requirements of the system. Currently, it is widely studied in software engineering and academia, but it is a relatively new research area in the automotive industry. Automotive cybersecurity testing usually relies on physical systems and is often performed at a later stage of the V-model development. Vulnerabilities identified later may require much more time and effort to fix. Therefore, model-based security testing shows certain advantages and can obtain security test cases according to the functional model or threat model of the system in the early stage of development and conduct security verification.

Cheah et al. [71] presented a test case generation method based on attack trees model using communicating sequential processes (CSP). CSP is a process-algebraic formalism for analyzing and modeling dynamic systems. They conduct testing on the Bluetooth and CAN. Since the construction of attack trees still mainly relies on manual effort, this is a semi-automatic approach. Heneghan et al. [72] further propose a framework for automated security testing of ECU at the component level based on CSP. They expect to identify vulnerabilities and verify functional security with model checking techniques. Mahmood et al. [73] provide a systematic MBST approach based on their work in [4], they design a software tool and a testbed for generating and executing test cases automatically. They launch several simulated attacks against the automotive Over-the-Air (OTA) system using the Uptane framework. Although only one type of attack is described in this paper, the effectiveness and prospect of this method are shown from the complete implementation process. Dos Santos et al. [74] considered the security flaws of automotive systems and vehicular network at an abstract level with Predicate/Transition (PrT) nets, which are a graphical dynamic system modeling language. They model four common attacks (interception, fabrication, modification of data, and interruption) and demonstrate the accuracy of the threat model in three real-world vehicle scenarios. They believe that the functional model of unified modeling language (UML) can be combined to generate the code of security test cases, but it was not implemented at the time.

Sommer et al. [75] proposed a concept of a security testing model, which is based on the vulnerabilities and attack privileges of the E/E architecture. They believe that the Extended Finite State Machine (EFSM) security model can be automatically generated through a formal description and point out that it is possible to generate test cases through model-checking techniques. Aouadi et al. [76] designed an automatic formal testing tool for distributed systems. They improved the tool by developing a user interface and also proposed a method to automatically generate test objects, which saves time and increases efficiency.

Marksteiner et al. [77] presented an approach to create a cyber digital twin model using binary analysis and generate test cases through formal transformation, model checking, and fault injection without a priori knowledge. In the same year, they also proposed the use of fingerprinting and model learning to construct attack tree models and utilize graph theory to generate attack paths. These approaches offer the possibility of automating cybersecurity testing, but these approaches are still at a conceptual stage [78]. Mahmood et al. [73] introduced an automated security testing approach, which uses attack trees for threat modeling. The model can then be formalized using CSP, and test cases can be automatically generated using model-checking techniques. Automated tools and security testbeds were developed to support the research. They performed an attack on the OTA update system and the experimental results demonstrate the effectiveness of their proposed approach.

Table 10 compares model types, model characteristics, and use cases. In addition, we distinguish whether the proposed method is a concept or a practical solution. Figure 10 depicts the distribution of papers by model type. The literature number on the attack tree model is the largest because the expression of tree models is intuitive and concise.

## 4. Automotive Cybersecurity Testing Testbeds

In this chapter, we address question RQ2 and conduct research on automotive cybersecurity testing testbeds. Unexpected situations can occur during security testing on an actual vehicle. Unrecoverable crashes or unintended activation of automobiles can cause economic and safety issues. Therefore, it is necessary to conduct testing on non-real, simulated automotive cybersecurity testbeds, which can save time and cost.

The testbed can simulate automotive network communication in a cost-effective way. Network communication includes in-vehicle communication such as CAN, LIN, and automotive Ethernet and out-of-vehicle communication such as Wi-Fi, cellular network, etc. In addition, testbeds can be divided into two types depending on whether they contain actual physical components: software and hybrid. Software-based testbeds enable virtual communication networks and ECU in the form of software, with no actual physical components in the system. This testbed is free from hardware constraints and is more convenient and flexible to use, but the test results may deviate from the actual system. Hybrid testbeds provide physical signals with physical components and can be controlled and monitored via software. It is a hardware-in-the-loop testbed that combines the best of both software and hardware.

Table 11 presents various automotive cybersecurity testbeds from 2010 to 2021. We briefly introduce their functionality and compare their characteristics from several aspects, including supported network protocols, types of testbeds, etc.

## 5. Discussion

This part will discuss the challenges and future trends of automotive cybersecurity testing in relation to RQ3.

We analyzed various security testing methods individually in the above. However, the security threats and vulnerabilities discovered individually in practical engineering may be limited. We need to integrate multiple testing methods into the continuous testing activities of automotive engineering. Designing a feasible test evaluation framework to assess the security risks of automobiles reliably is a topic that needs to be considered for a long time. Currently, several researchers have proposed some preliminary security testing frameworks. Strandberg et al. [91] proposed a theoretical SPMT methodology to predict and identify threats in the vehicle. Wooderson [9] introduced a framework for implementing cybersecurity engineering and described the tests suitable for different phases based on SAE J3061. Marksteiner [18] proposed a security testing framework that conforms to the SAE/ISO 21434 standard. This testing framework aims to prioritize risks and develop test scenarios for different risks to generate test cases. It is a good solution but has not been implemented yet. Ekert et al. [92] presented the verification and validation process framework required by 21434. Most of these frameworks have been proposed as concepts, and a few have been implemented as prototypes. It can be seen that the approach based on 21434 is becoming a trend, but whether or not it works needs to be verified in actual engineering.

While the standards and regulations for automotive cybersecurity have been released, there is no description of the technical details of security testing. There is no unified standard for researchers and testers to use for practical testing. Currently, security testing is not automated to a high degree, especially in penetration testing. Penetration testing may only utilize simple automated scripts and tools at certain stages, while manual effort is still required at other stages.

Currently, model-based approaches are still in the academic research phase, as this requires a high level of expertise, and if the system is particularly complex, building the model can also be challenging. Some studies have shown that there are still some problems in model-based testing, such as redundancy of test cases and optimization of models [93,94]. Some scholars propose a formalized model and apply appropriate coverage criteria to generate test cases, which provide a reference for automated testing [95]. Model-based testing technology is expected to automate testing in the future when modeling methods and tools mature.

In addition to model-based approaches, research on other advanced technologies, such as blockchain and artificial intelligence (AI), is also gradually increasing in the automotive field. Blockchain is regarded as a secure, trusted, and decentralized solution. Javed et al. [96] provided a comprehensive survey of blockchain and federated learning applications on vehicular networks. Jabbar et al. [97] compared the application of blockchain technology in intelligent transportation system. Kapassa et al. [98] discussed the limitations and contributions of blockchain technology in the application of the Internet of Vehicles. Zhou et al. [99] studied blockchain technology for secure authentication and trading between automobiles. AI is mainly applied in automotive security countermeasures, including secure communication [100,101], access control mechanisms [102], and IDS [103]. These two technologies are currently mainly utilized in security design, which can protect automotive cybersecurity, but their application in security testing remains to be studied.

## 6. Conclusions

Modern vehicles are connected and open systems. The attack surface and security risks to the vehicle have increased significantly. It is necessary to identify threats and vulnerabilities in the vehicle, and security testing plays a crucial role in this regard. Meanwhile, security testing is also an inevitable phase in the automotive development life cycle. With the release of ISO/SAE 21434 and WP.29 regulations, security testing of vehicles has become mandatory for type approval.

In this paper, we completed a comprehensive review of automotive cybersecurity testing. We performed statistics and classification of the screened literature. We classified automotive cybersecurity testing from three perspectives: knowledge level, level of automation, and test objective. Security testing can be divided into black-box, white-box, and grey-box types based on knowledge level. From the perspective of test tools and framework automation, testing can be classified into fully automated, semi-automated, and manual testing. For testing objectives, we focused on threat-based and requirement-based approaches, including vulnerability scanning, penetration testing, fuzzing, risk-based security testing, and functional security testing. Then, advanced model-based testing techniques were specifically described. Based on the surveyed literature, we compared the characteristics and applications of various methods. In addition, we also presented the automotive security testbeds between 2010 and 2021, which aims to provide a reference for researchers and engineers to study and establish an automotive security test platform. We introduced their functionality and compared their characteristics from protocols, types, and cost. Challenges still exist. We pointed out the limitations of current security testing frameworks and model-based automated testing but are optimistic about their future. Besides model-based testing, we also briefly outlined the application of blockchain and artificial intelligence technology in automotive cybersecurity. These two techniques are mainly applied in security countermeasures, such as authentication and intrusion detection, but their application in security testing remains to be studied.

In the future, we plan to work on how to apply these testing methods to actual usage scenarios in automotive engineering. Automation of security testing tools is also an important task. Moreover, by combining automation tools and model-based techniques, we expect to implement an automated automotive cybersecurity testing framework.

## Figures and Tables

**Figure 1 sensors-22-09211-f001:**
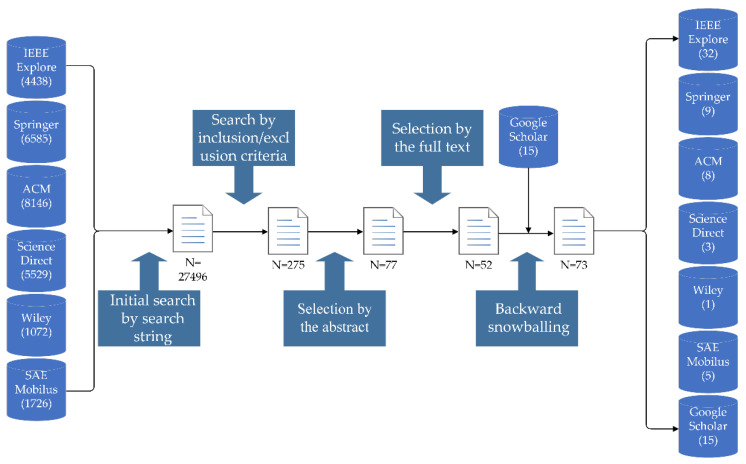
The process of literature search.

**Figure 2 sensors-22-09211-f002:**
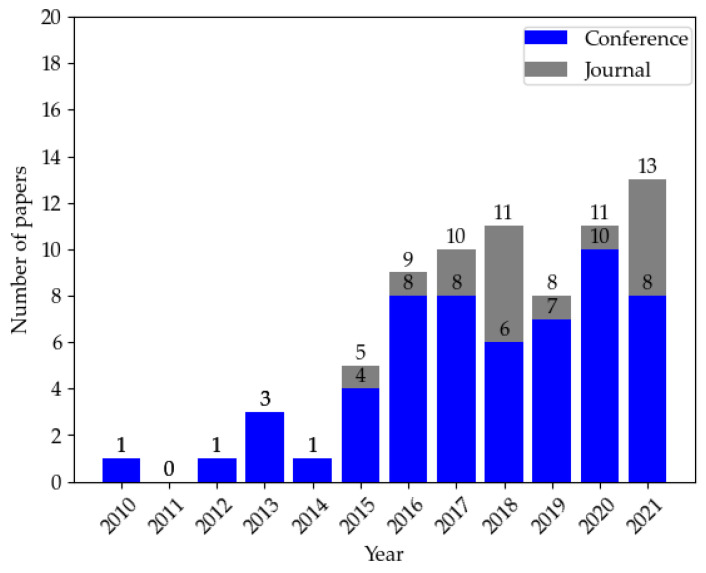
Distribution of papers by source from 2010 to 2021.

**Figure 3 sensors-22-09211-f003:**
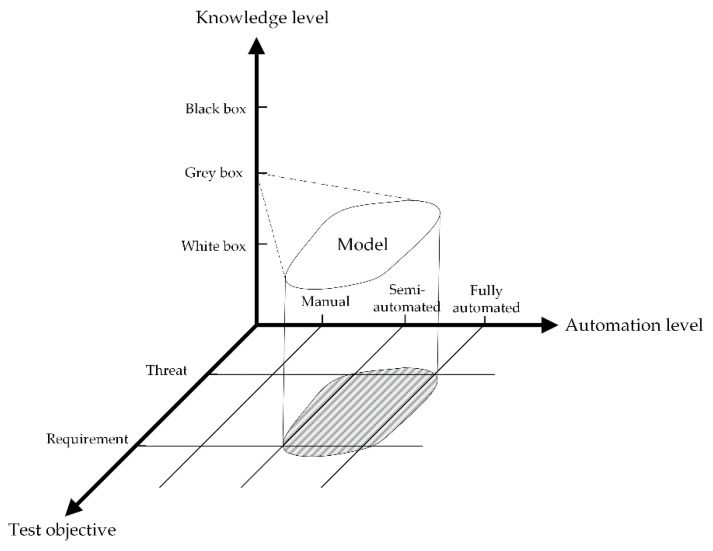
The classification of automotive cybersecurity testing methods.

**Figure 4 sensors-22-09211-f004:**
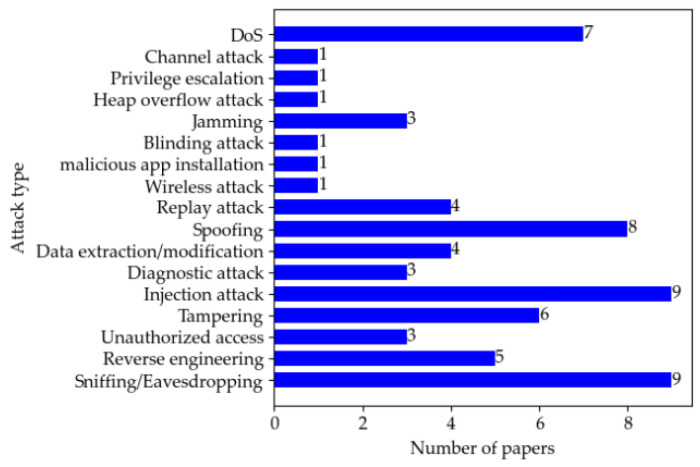
Distribution of papers by attack type.

**Figure 5 sensors-22-09211-f005:**
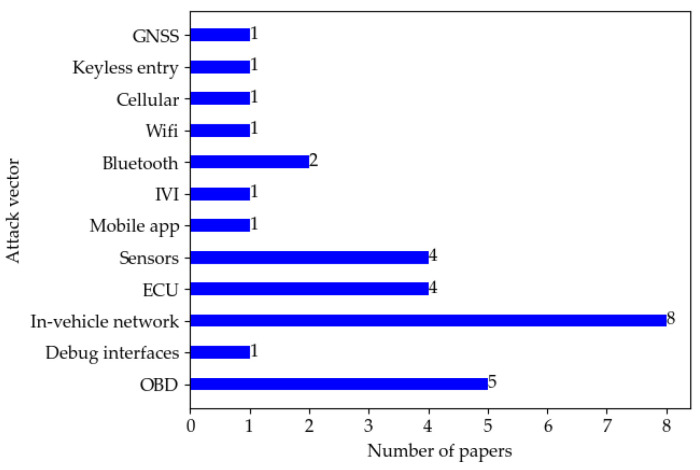
Distribution of papers by attack vector.

**Figure 6 sensors-22-09211-f006:**
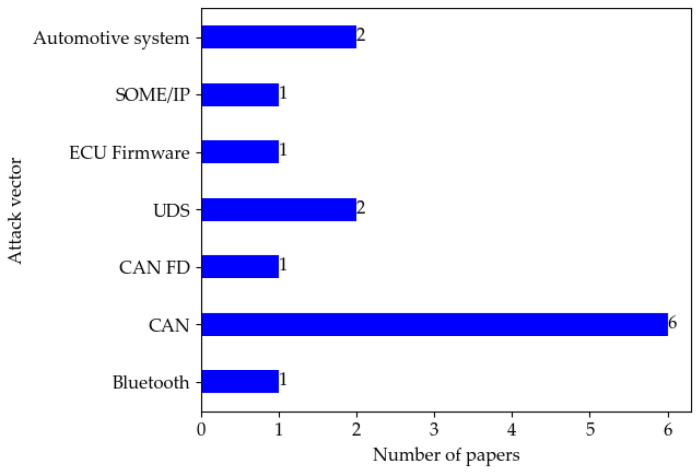
Distribution of papers by attack vector in fuzzing literature.

**Figure 7 sensors-22-09211-f007:**
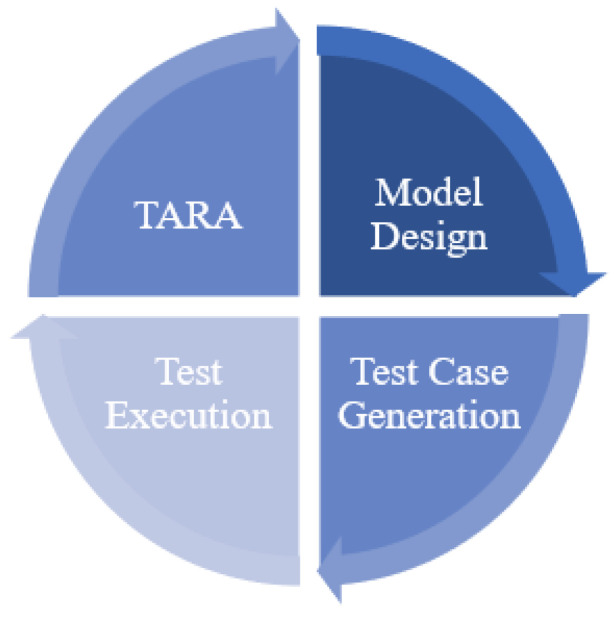
Generic process of risk-based security testing.

**Figure 8 sensors-22-09211-f008:**
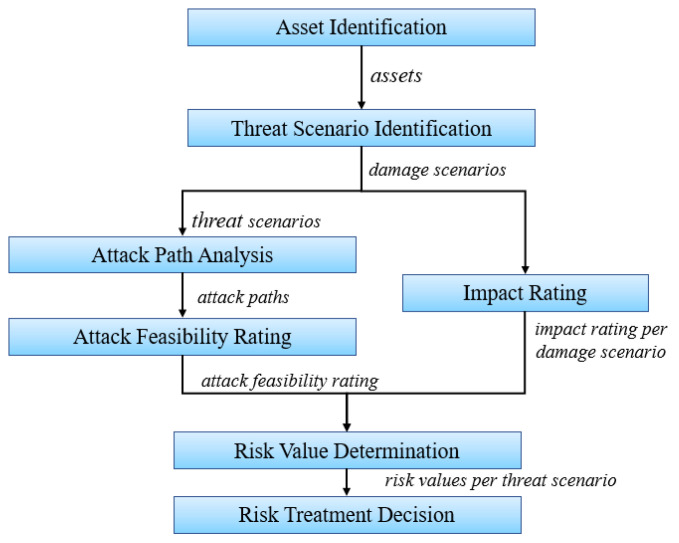
TARA in ISO/SAE 21434.

**Figure 9 sensors-22-09211-f009:**
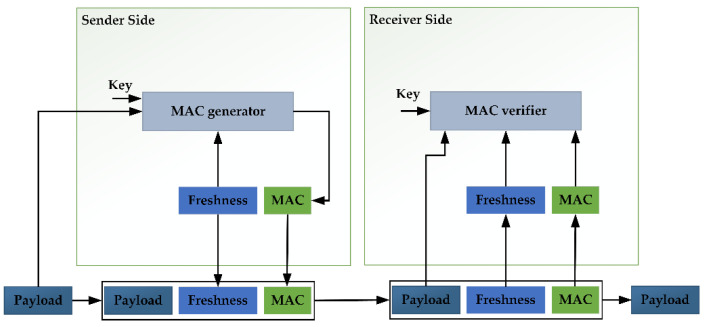
Freshness verification and message authentication process of SecOC [70].

**Figure 10 sensors-22-09211-f010:**
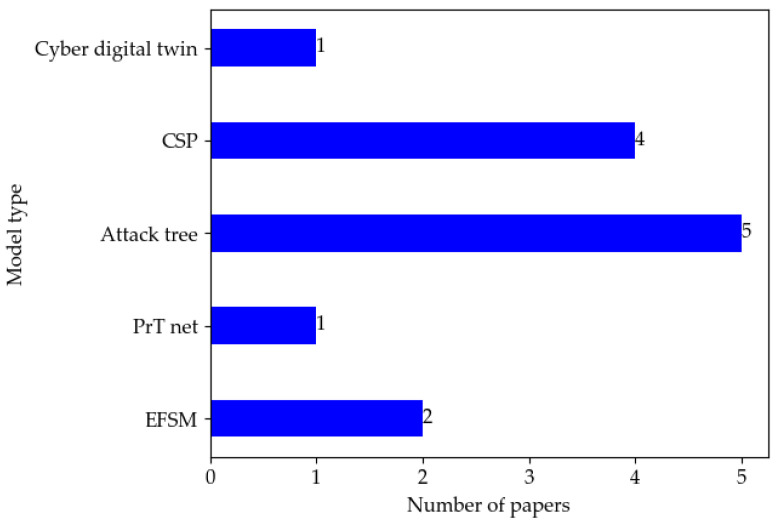
Distribution of papers by model type in the literature.

**Table 1 sensors-22-09211-t001:** Databases for searching.

Database	URL
IEEE Explore	https://ieeexplore.ieee.org/
Springer	https://link.springer.com/
ACM Digital Library	https://dl.acm.org/
ScienceDirect	https://www.sciencedirect.com/
Wiley	https://onlinelibrary.wiley.com/
SAE Mobilus	https://saemobilus.sae.org/
Google Scholar	https://scholar.google.com/

**Table 2 sensors-22-09211-t002:** Inclusion and exclusion criteria.

Aspect	Inclusion Criteria	Exclusion Criteria
Time	2010–2021	Not within the defined time
Language	Papers written in English	Papers not written in English
Accessibility	Full text is available	Full text is not available
Topic	Topics about security testing in the automotive domain	Papers related to general automotive security topics, including security mechanisms, design, etc.
Is it peer reviewed?	Yes	No

**Table 3 sensors-22-09211-t003:** Paper numbers in the search process.

Database	Initial Search	Inclusion/ Exclusion Criteria	Abstract	Full Text	Backward Snowballing
IEEE Explore	4438	81	42	30	32
Springer	6585	40	13	8	9
ACM Digital Library	8146	69	10	6	8
ScienceDirect	5529	52	3	2	3
Wiley	1072	16	1	1	1
SAE Mobilus	1726	17	8	5	5
Google Scholar	N/A	N/A	N/A	N/A	15
Total	27,496	275	77	52	73

N/A = Not applicable.

**Table 4 sensors-22-09211-t004:** Comparison of the knowledge-based testing.

Aspect	Black-Box Testing	Grey-Box Testing	White-Box Testing
Time	Short	Medium	Long
Cost	Low	Medium	High
Test coverage	Low	Medium	High
Knowledge of the target	Little	Medium	Much

**Table 5 sensors-22-09211-t005:** Comparison of the automated testing methods.

Aspect	Fully Automated Testing	Semi-Automated Testing	Manual Testing
Time	Short	Medium	Long
Cost	Low	Medium	High
Efficiency	High	Medium	Low
Knowledge of the target	Little	Medium	Much

**Table 6 sensors-22-09211-t006:** Comparison of various vulnerability scanning methods.

Aspect	Static/Dynamic Code Analysis	Communication Service Scanning	
		Wired	Wireless
Time	Long	Medium	Low
Cost	High	Medium	Low
Test scope	Application source code	Wired communication (In-vehicle network)	Wireless communication (Bluetooth, Wi-Fi, cellular)
Limitation	Source code required	Physical contact required	Physical contact not required

**Table 7 sensors-22-09211-t007:** A comparative overview of penetration testing in the surveyed literature.

Literature	Year	Attack Vector	Attack Type	Knowledge Level
Koscher [26]	2010	OBD, CAN, ECU	Sniffing, DoS, reverse engineering, unauthorized access, ECU tampering, injection attack	■
Miller [30]	2013	OBD, CAN, ECU	Sniffing, DoS, injection attack, diagnostic attack, firmware extraction/modification, ECU tampering, detecting attacks	■
Shoukry [31]	2013	ABS wheel speed sensors	Spoofing, tampering, injection attack	■
Woo [32]	2015	OBD, CAN, Mobile app, Bluetooth	Sniffing, DoS, replay attack, wireless attack, malicious app installation	■
Petit [33]	2015	Sensors	Blinding attack, jamming attack, replay attack relay attack, spoofing	■
Abbott-McCune [34]	2016	OBD, CAN	Sniffing, DoS, replay attack	■
Mazloom [27]	2016	IVI	Data extraction, reverse engineering, heap overflow attack, malicious code injection	■
Yan [28]	2016	Sensors	Jamming attack, spoofing	■
Nie [35]	2017	Wi-Fi, cellular, CAN	Privilege escalation, Unauthorized access, ECU tampering, reverse engineering, injection attack	■
Cheah [36]	2017	Bluetooth	Sniffing, DoS, data extraction, injection attack	■
Shin [29]	2017	Lidar	Channel attack	■
Milburn [37]	2018	CAN debug interfaces, ECU	Firmware extraction/modification, fault injection	■
Jeong [38]	2018	Keyless entry	Relay attack	■
Dürrwang [39]	2018	Airbag ECU	Diagnostic attack, signal tampering	■
Sommer [40]	2019	In-vehicle network	Eavesdropping, reverse engineering	■
He [41]	2020	GNSS	Spoofing, jamming attack	■
Zachos [42]	2020	OBD, CAN	Spoofing, diagnostic attack	■
He [43]	2020	OTA	Sniffing, DoS, spoofing, tampering, replay attack, unauthorized access, reverse engineering	■
Wen [44]	2020	Wireless OBD dongle	Spoofing, eavesdropping, injection attack	■
Ebert [16]	2021	Ethernet, IVI	DoS, spoofing, eavesdropping, malicious code injection	🞕

■ = Black box, 🞕 = Grey box.

**Table 8 sensors-22-09211-t008:** Overview of fuzzing in literature by different dimensions.

Literature	Attack Vector	Characteristics	Knowledge Level	Type	Testing Platform
Lee [45]	Bluetooth, CAN	Attacking ECU	■	Mutation	Instruments, real ECU
Fowler [46]	CAN	Reversing engineer, attacking network	■	Mutation	Vehicle simulator, an instrument cluster
Fowler [47]	CAN	Reversing engineer, inject message into ECU	■	Mutation	Display ECU
Werquin [48]	CAN	Reverse engineering	■	Mutation	Instrument Clusters
Radu [49]	CAN, ECU Firmware	Control flow graph, static analysis	☐	Generation	Real ECU
Zhang [50]	CAN	Bit flip rate, generative adversarial network	■	Hybrid	Actual vehicle, IDS
Nishimura [51]	CAN FD	Adaption for CAN FD, test execution time measurement	■	Mutation	Real ECU
Li [52]	SOME/IP	Attach fuzzing mode, structural mutation	🞕	Hybrid	Program from GENIVI/vsomeip
Bayer [53]	UDS	UDS fuzzing	☐	Generation	Simulated ECU
Patki [54]	UDS	UDS fuzzing	☐	Generation	Real ECU
Moukahal [55]	Automotive system	Vulnerability-oriented fuzz, structure-aware mutation	🞕	Hybrid	OpenPilot
Moukahal [56]	Automotive system	Prioritized and targeted concolic execution	🞕	Hybrid	OpenPilot

Hybrid = Generation + Mutation, ■ = Black box, ☐ = White box, 🞕 = Grey box.

**Table 9 sensors-22-09211-t009:** Overview of TARA methods in automotive domain.

Method	Brief Description	Application Scope	Threat Model	Co-Analysis
EVITA [59]	A method in the E-safety Vehicle Intrusion Protected Applications (EVITA) project which provides four evaluation dimensions: safety, privacy, financial, and operational.	Vehicular IT systems	Attack tree	Yes
HEAVENS [60]	A method in the HEAling vulnerabilities to enhance software (HEAVENS) project, which provides a complete evaluation process to propose a systematic approach so that cybersecurity requirements for automotive electrical and electronic systems can be obtained	Automotive Electrical and electronic systems	STRIDE	Yes
FMVEA [61]	FMVEA (Failure Mode, Vulnerabilities and Effects Analysis) extends the FMEA approach with security threat models	Automotive cyber-physical systems	STRIDE	Yes
SAHARA [62]	SAHARA (security-aware hazard analysis and risk assessment) is a method that combines HARA in functional safety and STRIDE threat models	Automotive embedded systems	STRIDE	Yes
SARA [63]	SARA is a systematic TARA framework that includes improved threat models, asset maps, new attack methods, attacker participation in the attack tree, and new driving system metrics	Automated driving system	STRIDELC, Attack tree, Attack map	Yes
CHASSIS [64]	CHASSIS (Combined harm assessment of safety and security) is a safety and security co-analysis method for information systems based on HAZOP guidewords	Automotive cyber-physical systems	HAZOP	Yes
TVRA [65]	TVRA (Threat, Vulnerabilities, and implementation Risks Analysis) is a process-driven threat analysis and risk assessment method proposed by the European Telecommunications Standards Institute (ETSI)	Automotive data/telecommunications networks	Threat tree	No
SINA [66]	SINA (Security in Networked Automotive) is a method to identify security issues for Connected automotive systems	Connected automotive systems	STRIDE, Attack tree	Yes
SGM [67]	SGM (Security Guideword Method) is a safety analysis method using security guide words	Automotive embedded systems	SGM, Attack Tree	Yes

STRIDE(LC) = Spoofing, Tampering, Repudiation, Information Disclosure, Denial of Service, Elevation of Privilege (Linkability, Confusion).

**Table 10 sensors-22-09211-t010:** Comparison of model-based testing approaches.

Author	Model Type	Model Characteristics	Use Case	Solution or Concept
Aouadi [76]	EFSM	An automatic formal testing tool for distributed systems is proposed, which permits the automatic generation of test cases	ITS	Solution
Dos Santos [74]	PrT net	Create threat and attack model with PrT net at an abstract level	Vehicular systems and networks.	Concept
Dos Santos [79]	Attack tree, CSP	Use CSP to create automotive bus systems and corresponding attack models	Vehicular systems	Concept
Cheah [80]	Attack tree, CSP	The attack tree can be transferred into a formal structure with CSP, and test cases can be generated automatically	Bluetooth-enabled OBD device	Concept
Cheah [71]	Attack tree	Develop a proof-of-concept tool to execute testing on vehicle compromise based on the attack model	Bluetooth, diagnostics device	Concept
Heneghan [72]	CSP	Security evaluation of ECU with CSP formal models	ECUs	Concept
Mahmood [73]	Attack tree, CSP	Construct threat model with attack trees and generate test cases by model-checking	OTA	Solution
Sommer [75]	EFSM	Security model with attack privileges and vulnerabilities	Vehicle networks	Concept
Marksteiner [77]	Cyber digital twin model	A cyber digital twin model using binary analysis and generating test cases through formal transformation, model checking, and fault injection	Vehicular systems	Concept
Marksteiner [78]	Attack tree	Construct attack tree model with fingerprinting and model learning	Vehicular systems	Concept

**Table 11 sensors-22-09211-t011:** Comparison of cybersecurity testbeds.

Testbeds	Network Protocol	Type	Cost	Functionality and Features
A cyber assurance testbed for heavy vehicle electronic controls [81]	J1939, J1708, CAN	H	Low	Supports remote experimentation, Key exchange strength and IDS assessment
A testbed for security analysis of modern vehicle systems [82]	CAN, Ethernet	H	Low	A testbed integrating CAN simulator and IVI system with flexible configuration for efficient security analysis
A testbed for automotive cybersecurity [83]	CAN	S	Low	A testbed consisting of a real-time CAN simulator and supporting reverse-engineering
ATG: An Attack Traffic Generation Tool for Security Testing of In-vehicle CAN Bus [84]	CAN	S	Low	Automatic generation of CAN attack traffic
PASTA: Portable Automotive Security Testbed with Adaptability [85]	CAN	H	Low	An open, safe, adaptable, and portable testbed against automotive attacks
A hardware-in-loop based testbed for automotive embedded systems cybersecurity evaluation [86]	V2X, CAN	H	Low	A hardware-in-the-loop testbed that can simulate V2X communication and perform GPS spoofing
A Connected Vehicle System (CVS) prototype testbed [87]	ETSI ITS-G5, 3G/4G, LTE, Ethernet, CAN, Wi-Fi	H	Medium	Support V2I communication test and security assurance evaluation
Ori: A Greybox Fuzzer for SOME/IP Protocols in Automotive Ethernet [52]	Ethernet, SOME/IP	S	Low	Fuzz SOME/IP protocols to find vulnerabilities
A novel testbed for automotive security analysis [88]	CAN	H	Medium	Data collected are analyzed visually and replayed through a test environment similar to a vehicle
SEPAD: Security Evaluation Platform for Autonomous Driving [89]	CAN/CAN FD, PLC, Bluetooth, WLAN, cellular, Ethernet	H	Medium	A novel testbed for autonomous driving, supporting security mechanisms testing
HybridgeCAN: a hybrid bridged CAN platform for automotive security testing [90]	CAN	H	Low	A low-cost testbed connecting physical ECUs with virtual components

S = Software, H = Hardware and software.

## Data Availability

Not applicable.

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
