# Peer review of "Cybersecurity Testing for Automotive Domain: A Survey"

_sensors, 2022, doi:10.3390/s22239211_

Round 1

Reviewer 1 Report

Summary: The authors of this paper propose a study of automobile cybersecurity testing research and practice. They categorize and discuss automotive engineering security testing methodologies and testbeds. They also indicate gaps and limitations in existing research and potential problems.

  Strong points: 1. The article is well-written and easy to read in general.
2. The covered topic is a hot topic of great importance. 3. The problem is well-motivated and well-defined. 4. The adopted methodology is clear and well organized 5. The references are appropriate.     Comments and Suggestions: 1. The main limitation of this work is that author are not covering articles published in 2022. The authors need to give a strong justification why they decided to ignore publications of this year.   2. More references should be included in the introduction by the authors. 

3. The authors could clarify the context of this research more thoroughly, especially why the research problem is relevant.

4. The introduction should explicitly highlight the major limitations of previous surveys similar to this paper.
  5. The authors may add a Figure at the begining of the article which illustrates the structure of the paper.   6. Section can be split into sub-sections   (i.e., a new subsection of every type of testing)   7. Google Scholar may be seen as a search engine and not a publisher of academic and scientific articles.   8. The author need to provide a more precise definition of the notion of "backward snowballing".   9. The key words used by authors for formulating their search query are very limited. The authors need to provide more justification about this choice.   10. One of the exclusion used by the authors criteria is: "Full text is not available". Does this mean that authors considered only open access articles?  

11. The authors ignored an important aspect related to the security of modern vehicles which corresponds to the use Blockchain Technology. The authors are invited to have a look on the following papers and to include them in their study: - https://www.mdpi.com/1424-8220/22/12/4394 - https://ieeexplore.ieee.org/abstract/document/9706476 - https://www.mdpi.com/1999-5903/13/12/313 - https://www.mdpi.com/1424-8220/22/22/8885 12. For section 3.5 related to "Model Based Testing", the authors need to mention the main challenges related to the application of this type of techniques and how these challenges may be tackled. For this purpose, the authors are invited to consider the following references which cover this issue: - https://ieeexplore.ieee.org/document/8859442 - https://www.sciencedirect.com/science/article/abs/pii/S0065245818300019 - https://link.springer.com/chapter/10.1007/978-3-319-94180-6_34 13. An other direction to be covered is related to the use of Artificial Inlelligence for cybersecurity purposes. The authors may consider the following references for clarifying this point: - https://ieeexplore.ieee.org/abstract/document/7943477 - https://ieeexplore.ieee.org/abstract/document/9474924 - https://www.hindawi.com/journals/scn/2022/3379843/ - https://iopscience.iop.org/article/10.1088/1757-899X/396/1/012017/meta

  14. The conclusion is too short and needs to be extended.  

Reviewer 2 Report

The submitted paper is an indeed thorough and up-to-date review on the latest advancements in the field of automotive cybersecurity. Its language is clear and correct. One especially outstanding point of the paper is the literature review methodology and its presentation.

Two points of improvement:

- Table 7. and 8. use custom (graphical) notation for knowledge level, which was not clear to me at first reading. (Legend is only provided at the bottom, and there was no mention of this specialty in the text.) What is more, Table 7. includes almost only black boxes (with only one exception at the bottom), so it was really unclear why those black boxes were there. Other readers may initially be confused too. (By reading further, this issue clears up since other “box versions” appear in Table 8.) Consider the usage of text instead of filled boxes or at least, explain it in advance!

- The presented techniques are well chosen and properly described, however, the paper lacks usage examples / scenarios where we could see these techniques applied as part of a complex testing setup. But this could be the subject of a next paper, since the current one is comprehensive even without it.

Round 2

Reviewer 1 Report

The authors have considered my comments and suggestions. I think this new version of the paper is ready for publication. Good Luck!